# CAD-Editor: Text-based CAD Editing through Adapting Large Language Models with Synthetic Data

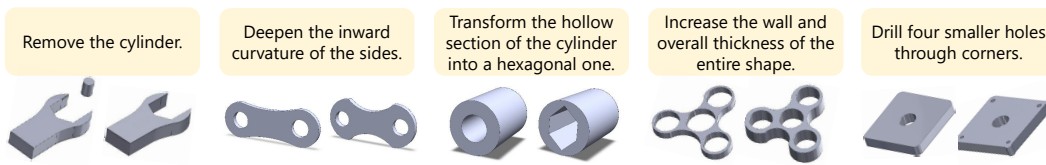

Figure 1: Text-based CAD Editing achieved by CAD-Editor. Each sub-figure shows the textual instruction at the top, the original CAD model on the left, and the edited CAD model on the right. The rendered image is shown for better comprehension. The actual editing occurs on sketch-and-extrusion (SE) operations of a CAD model to provide editability and reusability.

## Abstract

Computer Aided Design (CAD) is indispensable across various industries. *Text-based CAD editing*, which automatically modifies CAD models following textual instructions, is important yet not extensively studied. Existing work explores design variation generation, which randomly alters specific parts of a CAD model, offering no control over the final appearance. This work introduces *CAD-Editor* for text-based editing. We leverage Large Language Models (LLMs) as the backbone to take the concatenation of textual instruction and original CAD sequence as input and predict the edited CAD sequence, where the sequence representation of a CAD model is designed for easier processing by LLMs. Moreover, we propose fine-tuning LLMs by using a synthetic dataset followed by a selective dataset. The synthetic data is produced by leveraging powerful existing models, including design variation generation models for producing paired CAD models and multi-modal models for capturing textual differences between these pairs. The selective data is created by choosing top examples from outputs of the initially fine-tuned LLMs based on human feedback or metrics. In this way, a large-scale synthetic dataset offers basic capability while a selective dataset that is less noisy and better aligned with human intentions boosts performance further. Extensive experiments demonstrate the advantage of CAD-Editor both quantitatively and qualitatively.

## 1 Introduction

In the modern digital era, Computer Aided Design (CAD) has become indispensable across various industries, including automotive, aerospace, manufacturing, and architectural design. CAD is pivotal in creating everything from automobiles to airplanes and excavators to elevators, revolutionizing how we design and build. Most modern CAD design tools employ the "**Sketch-and-Extrude (SE) Operations**" (Shahin, 2008; Camba et al., 2016), where designers first sketch loops of 2D curves to define outer and inner boundaries of profiles, then extrude these profiles into 3D shapes, and finally combine these shapes to construct complex models. The creation of CAD models involves an iterative process, where an initial draft undergoes multiple modifications until it meets the user's needs. Throughout this procedure, natural language is widely used to convey how the CAD model should be adjusted. For those without design expertise, natural language offers the most accessible and straightforward means of expressing their needs. For professional designers, it acts as a vital

medium for fast, detailed, and clear communication. Consequently, a system capable of automatically editing a CAD model based on textual instructions, called **text-based CAD editing**, could revolutionize the entire CAD design process. It would greatly speed up CAD model development and enable more people, particularly those with limited design expertise, to create CAD models.

Text-based CAD editing presents several distinct challenges. First, the textual instructions are highly varied, such as deletion (the 1st case in Figure 1 ), addition (the 5th case in Figure 1), local change (the 2nd and the 3rd case in Figure 1) and global change (the 4th case in Figure 1). Besides, it demands on an accurate understanding of the structure of CAD's SE operations. For example, to complete the editing task in the 2nd case of Figure 1, one must identify which part of the SE operations represents the 'inward curvature'. Moreover, it also requires comprehending geometric concepts, such as specific shapes as the cylinder and hexagonal holes shown in the 1st and 3rd cases, and positions like corners shown in the 5th case of Figure 1. Additionally, there is an absence of naturally-existing datasets, making it even more intractable.

While text-based CAD editing is important and challenging, it receives little research attention. Existing work (Wu et al., 2021; Xu et al., 2022; 2023) primarily involves altering specific parts of a CAD model randomly while leaving others unchanged, without any control over the final appearance of the new CAD model, called *design variation generation* for brevity. For example, SkexGen (Xu et al., 2022) supports variations in a CAD model's topology, geometry, or extrusion level through disentangled codebooks. Hnc-CAD (Xu et al., 2023) enables variations at loop, profile, or solid levels and completes an entire CAD model from a partial one using a neural code tree representation.

In this work, we aim to exploit the capabilities of large-scale models and high-quality data, rather than exploring specialized methods, to tackle the mentioned challenges for the first time. This is motivated by the success of large language models (LLMs), where the model size and data quality have been shown to be key factors in their success (OpenAI, 2023; Touvron et al., 2023). Moreover, developing more specialized techniques depends on having a large enough model and sufficient training data as an initial step. Specifically, our main idea is to utilize pre-trained LLMs as the backbone, generate a large-scale synthetic dataset for initial fine-tuning, and enhance the fine-tuning with a small-scale selective dataset. There are three main advantages to this approach. First, LLMs inherently possess strong text comprehension abilities, which are beneficial in handling diverse and complex textual instructions. Second, there is evidence that LLMs have learned CAD-related codes in the pre-training (Makatura et al., 2023), potentially aiding in understanding CAD's SE operations and geometric concepts. Third, the combination of synthetic and selective data offers a practical and efficient way to obtain sufficient training data when naturally existing datasets are absent.

To achieve this, as an initial step, we formulate text-based CAD editing as a sequence-to-sequence (seq2seq) problem, where the input is the concatenation of the textual instruction and the sequence of the original CAD model and the output is the sequence of the edited CAD model. Here, the CAD sequence is constructed by adjusting the existing design of the CAD sequence Xu et al. (2022) to enhance their readability and comprehensibility for LLMs. Specifically, we allow for variable length and use textual tokens to represent both categorical and numerical variables in CAD's SE operations. Next, we propose synthesizing a high-quality training dataset by summarizing CAD model variations and using it for the initial fine-tuning of LLMs. Concretely, we harness the strong capability of existing models to efficiently synthesize this data. Existing design variation generation models are used to produce paired CAD models, and pre-trained Multi-modal LLMs (MLLMs) with a multi-level captioning strategy are employed to derive the textual differences between images rendered from these paired CAD models. Moreover, we construct a selective dataset with less noise and better alignment with human intentions and use it for further fine-tuning LLMs to improve performance. Specifically, this selective dataset is built on the observation that sampling the initially fine-tuned model several times will yield at least one good result. We use human feedback or a metric—directional CLIP score—to choose a small-scale set of good examples from multiple outputs of the initially fine-tuned LLM. We refer to the entire framework as **CAD-Editor** for ease of reference. The major contributions of this work are as follows:

- We introduce a new task called text-based CAD editing, which allows for precise edits via textual instructions, aligning more closely with real-world user needs.
- We propose CAD-Editor, which exploits the potential of large-scale models and high-quality data to develop the first model for text-based CAD editing. We formulate text-based editing as a seq2seq problem and employ pre-trained LLMs as our backbone. Moreover,

we present efficient and practical techniques to build both a synthetic dataset and a selective dataset, which are then used to fine-tune LLMs in a stepwise manner.

- We build the first benchmark for text-based CAD editing based on DeepCAD dataset (Wu et al., 2021), and demonstrate the superiority of CAD-Editor over baselines in terms of sequence validity, text-CAD alignment, and generation quality.

## 2 RELATED WORKS

**CAD Generation.** Parametric CAD, characterized by its sketch-and-extrude operations, dominates mechanical design. The recent development of large-scale parametric CAD datasets has facilitated the adoption of learning-based approaches. These methods utilize the historical sequence of CAD modeling and sketch constraints to generate engineering sketches and solid models (Willis et al., 2021b; Wu et al., 2021; Xu et al., 2022; 2023; Seff et al., 2020). The resulting sequences can be processed by a solid modeling kernel to produce editable parametric CAD files, which include either 2D engineering sketches (Willis et al., 2021a; Seff et al., 2022) or 3D CAD shapes (Xu et al., 2022; 2023). CAD generation models have focused on tasks such as reverse engineering from point clouds (Khan et al., 2024; Ma et al., 2024), unconditional generation (Wu et al., 2021), and design variations (Xu et al., 2022; 2023). For instance, SkexGen (Xu et al., 2022) allows fixing topology codes to maintain shape geometry, geometry codes to maintain dimensions and positioning, or extrusion codes to maintain the height of extruded sketches along with their 3D combinations. Similarly, Hnc-CAD (Xu et al., 2023) leverages the inherent hierarchies within CAD models to facilitate editing at the loop, profile, and solid levels, and offers auto-completion by adding or attaching additional sketches. While previous approaches can generate diverse shapes based on high-level guidance, they overlook the challenge of enabling CAD editing through textual instructions, which often results in arbitrary editing rather than meaningful changes. Providing user control over the generation process, while preserving design intent, is key for the adoption of generative models in real-world CAD software. Different from existing works, our method leverages both the sketch-and-extrude features and the natural language capabilities of pre-trained LLMs to enable instruction-based editing, which allows users to directly edit through explicit textual instructions.

**Text-based Editing.** Text-based editing is crucial across various domains, as textual instructions are expressive, precise, and intuitive, enabling users to easily isolate and modify specific objects or visual attributes. This task has been extensively explored in areas such as 3D editing (Mikaeili et al., 2023), image editing (Meng et al., 2021; Brooks et al., 2023), and video editing (Chai et al., 2023; Ceylan et al., 2023). For example, InstructPix2Pix (Brooks et al., 2023) utilizes synthetic data generated by a language model and a text-to-image model to create a large dataset of image editing examples, enabling instruction-based edits for images. StableVideo (Chai et al., 2023) introduces a text-driven video editing framework that employs a novel inter-frame propagation mechanism to achieve consistency-aware video editing. These advancements have inspired our approach to text-based CAD editing, and we are the first to apply text-based editing within the CAD domain.

**Large Language Models.** In recent years, scaling pre-trained language models (PLMs) has consistently enhanced their capacity for downstream tasks. This trend has led to the development of increasingly larger PLMs, such as ChatGPT OpenAI (2023), GPT-4 Achiam et al. (2023), and Gemini-Pro Team et al. (2023), each boasting over 100 billion parameters. These models, collectively known as large language models (LLMs), have gained widespread attention. The availability of open-source LLMs like LLaMA Touvron et al. (2023) has further accelerated research in this field. LLMs distinguish themselves from smaller models through remarkable emergent capabilities, particularly in-context learning Brown et al. (2020) and chain-of-thought prompting Wei et al. (2022); Kojima et al. (2022). They have revolutionized generative tasks, showing significant improvements over traditional neural networks in areas such as code generation Chen et al. (2021); Nijkamp et al. (2022); Chen et al. (2023) and material generation Gruver et al. (2024). Additionally, the strategic use of LLMs for generating synthetic data to aid training has opened new avenues for research and application Xu et al. (2024); Yu et al. (2024). This synthetic data is often rich and diverse, providing substantial benefits for model training and performance.

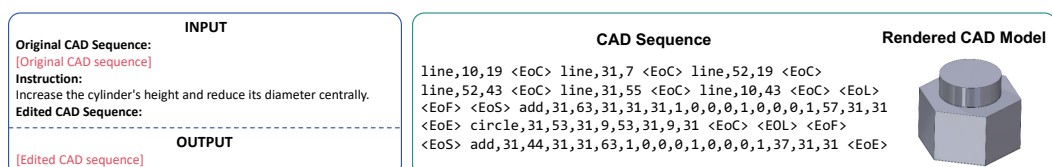

Figure 2: **Left:** Example input and output for CAD-Editor. The input combines the original CAD sequence with the textual instruction, and the output is the modified CAD sequence. The specific CAD sequence is shortened to '[Original (or Edited) CAD Sequence]' to save space. **Right:** An illustration for a specific CAD sequence and its rendered CAD model.

## 3 METHODS

In this section, we present CAD-Editor, the first generative model for text-based CAD editing. We frame the task as a seq2seq generation problem, where the sequence representation of a CAD model is adapted for easier process and comprehension by LLMs (Section 3.1). Then, we introduce a initial fine-tuning stage of LLMs on a synthetic dataset, which is produced by summarizing CAD model variations (Section 3.2). Moreover, we enhance this with further fine-tuning on a selective dataset, which is built by choosing top examples from outputs of the initially fine-tuned LLMs (Section 3.3).

### 3.1 TEXT-BASED CAD EDITING AS SEQ2SEQ GENERATION

In previous work, a CAD model has been formulated as a sequence for other CAD generation tasks (Wu et al., 2021; Xu et al., 2022; 2023). Building upon these efforts, we formulate the text-based CAD generation as a seq2seq generation task and adjust the sequence representation of a CAD model (Xu et al., 2022) to be more comprehensible for LLMs. As shown in the left side of Figure 2, we combine the textual instruction with the original CAD sequence as input, and represent the output as the edited CAD sequence. Moreover, as illustrated in the right side of Figure 2, SE operations of a CAD model are expressed as a series of text tokens. These include 1) topology tokens indicating a curve type (e.g., line, arc, or circle), 2) geometric tokens for point coordinates which are discretized into integers and then represented as text, 3) extrusion tokens where both Boolean operations (e.g., add, cut, or intersect) and numerical parameters are converted to text, and 4) end-primitive tokens (e.g., EoC, EoL, EoF, EoS or EoE for the end of a curve, loop, face, sketch or extrusion). Unlike previous work, we use text tokens (i.e., natural language descriptions) instead of binary representations for categorical operations such as curve types and extrusion operations as well as numerical attributes such as point coordinates and extrusion parameters. Besides, we support variable token lengths instead of using placeholders to standardize the command length. This approach simplifies processing and interpretation for LLMs and reduces the sequence length.

### 3.2 INITIAL FINE-TUNING WITH SYNTHETIC DATASET

With the seq2seq formulation introduced above, it is feasible to fine-tune pre-trained LLMs for text-based CAD editing, provided there is a training dataset comprising triplets of ⟨textual instruction, original CAD sequence, edited CAD sequence⟩. However, such data is neither naturally available nor cost-effective to manually label. Existing models, fortunately, offer potential solutions for constructing these triplets in an economical and efficient manner. Design variation generation models can generate paired CAD models but lack the ability to produce corresponding text instructions (Xu et al., 2022; 2023). On the other hand, multi-modal LLMs (MLLMs) cannot generate CAD models but excel in understanding rendered CAD images (Achiam et al., 2023). Therefore, we propose harnessing the complementary strengths of these existing models to systematically generate the needed training set. For ease of reference, we call this training set *synthetic dataset*, denoted by $\mathcal{D}_{\text{synthetic}}$. Below, we detail the two critical steps for generating this synthetic dataset $\mathcal{D}_{\text{synthetic}}$ (see Figure 3).

**Paired CAD Generation by Design Variation.** In this step, we get CAD pairs by feeding an existing CAD model into design variation generation models to obtain its variants, with certain parts of the CAD model being randomly altered. The process of CAD model editing naturally involves modifying, adding, and removing operations. To comprehensively capture these aspects, we select

Figure 3: **Left:** Initial fine-tuning on the synthetic dataset, which is produced by summarizing CAD model variations. **Right:** Enhanced fine-tuning on the selective dataset, which is built by choosing top examples from outputs of the initially fine-tuned LLMs.

suitable design variation models for constructing CAD pairs. Specifically, we generate CAD pairs related to modifying operations by using SkexGen (Xu et al., 2022), where a certain code in the codebook is fixed to achieve this. Besides, we generate CAD pairs related to adding operations by using Hnc-CAD (Xu et al., 2023), where auto-completion is utilized to achieve this. Additionally, we use the generated CAD model from Hnc-CAD's auto-completion as the source and the original CAD model as the target to construct the CAD pairs related to removing operations.

The design variation generation model can sometimes introduce noise by altering a CAD model too drastically or not at all. To ensure data quality, we implement a data filtering strategy. Specifically, we render each generated CAD pair into images, calculate the cosine similarity between them, and exclude pairs with similarity below a certain threshold. In our experiments, thresholds are set at $0.95$ for the CAD pairs generated by SkexGen and $0.9$ for Hnc-CAD. Additionally, we filter out CAD pairs showing no significant changes in the CAD sequence level.

**Textual Instruction Generation by Multi-level Captioning.** In this step, we render the CAD pairs from the last step as images and input the images into MLLMs to get the textual difference. Note that we opt to identify differences at the rendered image level instead of the CAD sequence level because changes are more easily perceived visually. In the preliminary study, we observe that even advanced MLLMs, e.g., GPT-4o, often make mistakes when directly describing how to edit one CAD model into another, such as incorrect positional relationships, numbers of shapes, or types of shapes. To mitigate this issue, we introduce a multi-level captioning strategy, which breaks down this complex task into several simpler sub-tasks, thereby improving the overall accuracy. Specifically, this method involves the following three sub-tasks, with full prompts available in Appendix A. 1) *Describe CAD images*. Initially, we generate detailed descriptions for each CAD image in the pair with MLLMs (e.g., GPT-4o). This step involves a meticulous examination of each image's geometric properties, including the type and number of elements, the proportions of sizes, positional relationships between elements, and any additional noteworthy details. 2) *Identify differences*. Next, we feed both the detailed descriptions and the CAD images into the MLLM to generate editing instructions, summarizing necessary modifications in a clear and direct manner. The detailed description helps in mitigating the complexity of identifying changes solely from the images. 3) *Compress instructions*. Lastly, we compress the editing instruction into a single sentence, limited to a maximum of 10 words. This ensures that the instructions are precise yet concise, making them easier to understand.

Similarly, we also implement data filtering to ensure data quality. We filter out CAD pairs with too many changes by excluding instructions with more than three edits. We also filter out pairs with no significant changes by excluding instructions containing phrases like "no transformation is needed".

Finally, by assembling the CAD pairs from the first step and the textual instruction from the second step, we derive a synthetic dataset $\mathcal{D}_{\text{synthetic}}$ with triplets of ⟨textual instruction, original CAD sequence, edited CAD sequence⟩. We then fine-tune pre-trained LLMs using Low-Rank Adapters (LoRA) (Hu et al., 2022) on this synthetic data, resulting in a basic model for text-based editing referred to as an *initially fine-tuned LLM*.

## 3.3 ENHANCED FINE-TUNING WITH SELECTIVE DATA

While Section 3.2 makes every effort to ensure the high quality of the synthetic data, achieving absolute accuracy is impossible. We seek to construct a *selective dataset*, denoted as $\mathcal{D}_{\text{selective}}$, which

Table 1: Quantitative evaluations on the text-based CAD editing task. SkexGen and Hnc-CAD are unable to handle text-based editing, so only their generation quality is compared. MMD, JSD, and D-CLIP values are multiplied by $10^2$. ↑: the higher the better, ↓: the lower the better.

| Method | COV ↑ | MMD ↓ | JSD ↓ | D-CLIP ↑ | Valid Ratio ↑ | Human Eval ↑ |
|---|---|---|---|---|---|---|
| SkexGen | 80.2 | 1.38 | 1.72 | - | 69.6 | - |
| Hnc-CAD | **81.9** | 1.27 | 1.68 | - | 78.6 | - |
| GPT-4o (basic) | 78.4 | 1.26 | 2.06 | - 0.65 | 45.5 | 7.09 |
| GPT-4o (3-shot) | 80.9 | 1.22 | 1.70 | - 0.01 | 77.0 | 19.8 |
| CAD-Editor | 81.0 | **1.16** | **1.55** | **0.21** | **91.4** | **31.1** |

contains less noise and aligns better with human intentions. Then, we further fine-tune the initially fine-tuned LLMs on this dataset to enhance the performance. Our method for constructing such a dataset hinges on a key observation: while the initially fine-tuned LLMs may not consistently generate satisfactory results, sampling multiple times often yields at least one exceptional outcome. The primary challenge is determining whether the model outcome is sufficiently high-quality. We propose two feasible methods: automatic selection using metrics and manual selection. Both methods have been shown to enhance performance, with the former being more efficient. We anticipate that better metrics will be developed in the future to achieve even better results witin our framework.

**Automatic Selection with Directional CLIP (D-CLIP) Score.** Directional CLIP (D-CLIP) is initially proposed to measure whether the changes between two images align with the changes between two texts in the CLIP space for image domain adaption problem (Gal et al., 2022). In this work, we adapt it to select top examples from model outputs. Specifically, after sampling multiple edited CAD sequences from the initially fine-tuned LLMs, we render them into CAD images. Then, we leverage D-CLIP to calculate how much the change between the image for the edited CAD model and the image for the original CAD model agrees with the textual instruction:

$$\Delta T = E_T\left(t_{\text{edited}}\right) - E_T\left(t_{\text{source}}\right), \quad \Delta I = E_I\left(i_{\text{edited}}\right) - E_I\left(i_{\text{source}}\right), \quad D\text{-}CLIP = \frac{\Delta I \cdot \Delta T}{|\Delta I||\Delta T|}.$$

$E_I$ and $E_T$ are CLIP's image and text encoders, $t_{\text{source}}$ is a neutral text (e.g.,"This is a 3D shape."), $t_{\text{edited}}$ is the concatenation of $t_{\text{source}}$ and the textual instruction. $i_{\text{edited}}$ and $i_{\text{source}}$ are the images for the edited and original CAD model. We select the edited CAD model with the highest D-CLIP and pair it with the original CAD model and textual instruction to form a triplet, which is then added to in the selective dataset. We refer to this method as *D-CLIP feedback (DCF)* for brevity.

**Manual Selection.** Similarly, we render the edited CAD sequence generated by the initially fine-tuned LLMs into images. Then, we let humans select which one is the best. The chosen one, together with the corresponding original CAD model and the textual instruction is added into the selective dataset. We refer to this method as *human feedback (HF)* for brevity. Compared to DCF, HF is a more direct way for injecting human intention into the model training.

## 4 EXPERIMENTS

### 4.1 EXPERIMENTAL SETUP

**Datasets.** We use the DeepCAD dataset (Wu et al., 2021), which contains 178,238 CAD models split into 90% training, 5% validation, and 5% testing segments. We utilize the same strategy in existing work (Xu et al., 2022; 2023) to remove duplication, and exclude the long-tail data with more than 3 sketch-extrusion pairs and 20 curves. For the synthetic dataset used for training, we get 92,070 examples by following the method in Section 3.2,. For the test set, we randomly sample 2000 examples from the original test segment, get the initial version following Section 3.2, and manually examine them to ensure the correctness. To compare the performance of different methods, we generate 5 outputs for each example in the test set, yielding 10,000 CAD models for evaluation.

**Metrics.** For quantitative comparison with 3D generative models, we utilize metrics from previous works (Wu et al., 2021; Xu et al., 2022; 2023). "Coverage" (COV) represents the percentage of

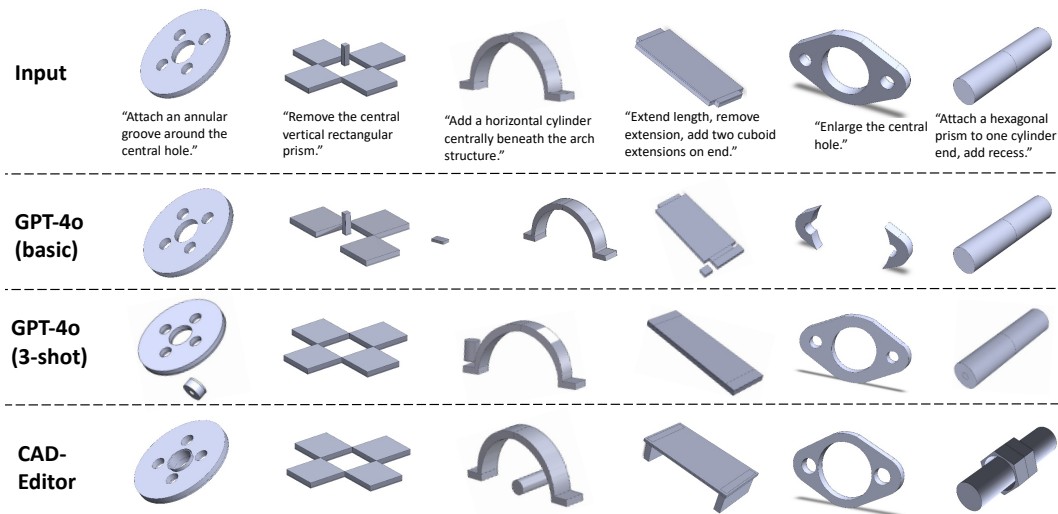

Figure 4: Results for text-based CAD editing by GPT-4o (basic), GPT-4o (3-shot) and CAD-Editor.

real data that matches generated data based on the closest Chamfer distance of uniformly sampled points on the surface. "Minimum Matching Distance" (MMD) is the average minimum matching distance between a generated sample and its nearest neighbor in the real set. "Jensen-Shannon Divergence" (JSD) measures the similarity between real and generated distributions based on the marginal point distribution. Additionally, we use D-CLIP, as described in Section 3.3, to assess how well the changes in CAD models align with the editing instructions at the image level. It is important to note that the output CAD command sequence does not always produce a valid 3D shape. In some cases, the output commands may result in an invalid curve or other issues, making it impossible to extract a point cloud from that CAD model. Therefore, we also report the Valid Ratio, which is the percentage of output CAD models successfully rendered to an image.

**Implementation Details.** We use Llama-3-8b-Instruct as our base LLM, fine-tuning it for 70 epochs using PyTorch's Distributed Data Parallel (DDP) on 4 NVIDIA A6000-46GB SMX GPUs. The initial learning rate is set to 1e-4 with a maximum token length of 1024. We employ Low-Rank Adapters (LoRA) with a rank of 32. When employing the LLM for inference, we set the temperature as 0.9 and top-p as 0.9 to generate varied results in each trial.

**Baselines.** In our experiment, we compare our results with baseline methods, including: 1) previous CAD design variation generation models such as SkexGen and Hnc-CAD, and 2) foundation models not specifically tailored for CAD generation tasks, including closed-source commercial LLMs like GPT-4o. We design two prompting methods: the first is a basic method that explains the design rules of CAD operation sequences; the second is a dynamic 3-shot method, which, in addition to the basic explanation, includes the three most similar instructions based on text cosine similarity and their corresponding CAD pairs from the training set as in-context learning examples. The detailed prompt can be found in Appendix B.

## 4.2 MAIN RESULTS

**Quantitative Evaluation.** Table 1 reports the average scores across 3 runs. Notably, CAD-Editor achieves a high Valid Ratio of 91.4%, significantly surpassing other methods and indicating a greater proportion of valid and high-quality CAD generations. In terms of D-CLIP, which measures alignment with editing instructions, CAD-Editor scores 0.21, a substantial improvement over GPT-4o (basic) at -0.65 and GPT-4o (3-shot) at -0.01. This underscores CAD-Editor's effectiveness in adhering to user instructions. Additionally, CAD-Editor outperforms GPT-4o on all three point cloud evaluation metrics (COV, MMD, and JSD) and performs comparably to SkexGen and Hnc-CAD, demonstrating exceptional quality and diversity in its CAD edits. Overall, the results indicate that

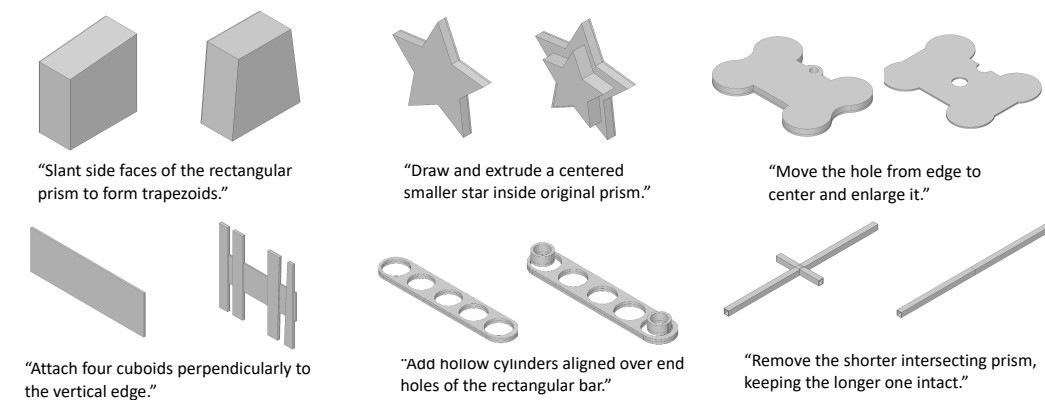

Figure 5: More results with various textual instructions from CAD-Editor.

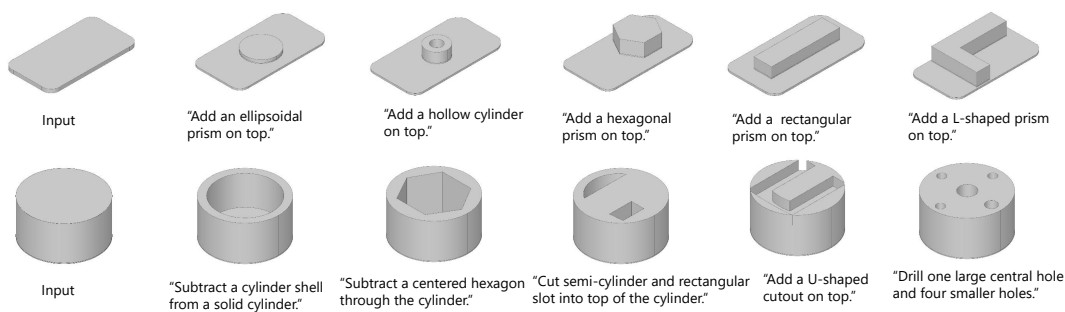

Figure 6: Given one CAD model and various instructions, CAD-Editor produces different outcomes.

CAD-Editor not only enhances the quality and diversity of CAD edits but also ensures better alignment with user instructions and higher validity of the generated designs.

**Qualitative Evaluation.** In Figure 4, we qualitatively compare our method with GPT-4o (basic) and GPT-4o (3-shot). We observe that GPT-4o (basic) often generates irrelevant edits (case 4), unrealistic shapes (case 5), or fails to make any changes (cases 1 and 6). Additionally, it struggles with distinguishing shape types (case 3) and locating specific positions (case 2). It performs better with dynamic few-shot prompting, highlighting the quality of our synthetic data. GPT-4o (3-shot) can detect specific shapes reasonably well but still struggles with precise localization (case 3). In contrast, our model successfully executes many challenging edits, including modifying sizes, shapes, and positions, as well as replacing, adding, and removing objects. Figures 5, 6, 7, and 8 show more selected results.

**Human Evaluation.** We randomly sampled 2,500 CAD models from the entire set of generated results. Each image pair was independently rated by five crowd workers. For each pair, a score of 1 was assigned if the generated data was deemed successful, and 0 otherwise. Success was defined by two criteria: alignment with the text and sufficiently high quality. The results are in Table 1. The Human Evaluation score (Human Eval) of 31.1% underscores the enhanced user satisfaction and editing quality of CAD-Editor. Our method achieved higher scores compared to GPT-4o (both basic and 3-shot prompt settings). This indicates that crowd workers frequently identified models generated by GPT-4o as failing to follow the editing instructions, whereas our method demonstrated superior performance.

### 4.3 ABLATION STUDIES

**Multi-level Captioning.** A straightforward way to generate the textual difference between CAD model pairs in Section 3.2 is to directly query MLLMs (e.g., GPT-4o). We denote this method

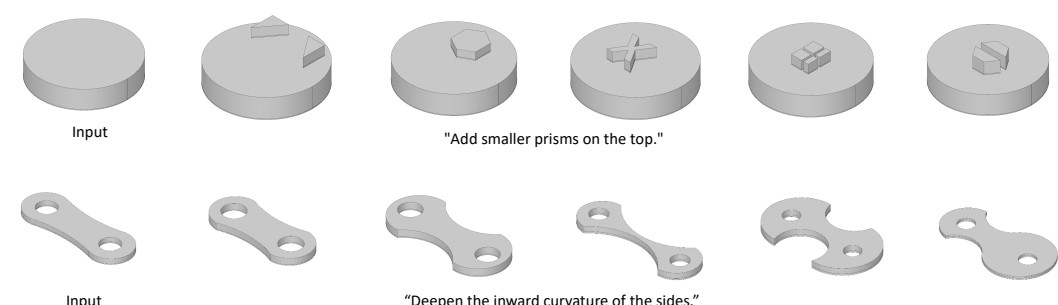

Figure 7: Given the same CAD model and instruction, CAD-Editor produces diverse outcomes.

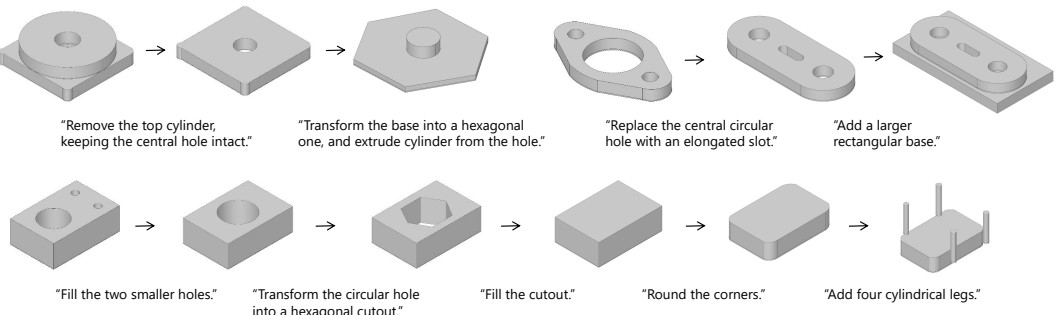

Figure 8: Apply CAD-Editor iteratively to edit a CAD model until it meets user requirements.

as *basic captioning*. In Figure 9, we present qualitative ablations comparing the basic captioning with our multi-level captioning. The basic captioning often fails to accurately capture detailed positional relationships, numbers, and shape types, resulting in imprecise and sometimes erroneous captions. This can bring too much noise to the training data. In contrast, our multi-level captioning decomposes the captioning task into smaller, more manageable sub-tasks. This hierarchical approach allows for a more precise capture of intricate details and relationships within the CAD models. Moreover, we make a quantitative comparison. Given the restricted computing resources, we sample 10,000 examples from the entire synthetic training set, and obtain basic captions as well as multi-level captions for these samples. We fine-tune LLaMA-3-8B on these datasets using the same settings as described in Section 4.1. The results, detailed in Table 3, indicate that multi-level captioning (CAD-Editor-mini w/MLC) leads to better performance compared to basic captioning (CAD-Editor-mini w/BC). Specifically, the quality of the output improves significantly, with higher COV, lower MMD and JSD, better adherence to user instructions as measured by D-CLIP, and a higher Valid Ratio.

**Selective Data.** We compare three settings: no enhanced fine-tuning with selective data, using DCF to collect selective data and using HF to collect selective data, denoted as CAD-Editor w/o DCF&HF, CAD-Editor w/DCF, and CAD-Editor w/ HF in Table 3. Compared to not using selective data, using DCF and HF significantly improves alignment between textual instruction and edited CAD model (measured by D-CLIP). More qualitative comparisons are included in the Appendix.

## 5 CONCLUSION

We introduced CAD-Editor, a novel generative model marking a significant advancement in the field of CAD, particularly in the under-explored area of text-based CAD editing. By designing a sequence representation of CAD models suitable for LLM processing and employing a two-stage fine-tuning process with synthetic and selective datasets, CAD-Editor achieved compelling performance. This is a step towards an intelligent system capable of generating diverse CAD models that align with user intentions. However, CAD-Editor is limited by the quality of the generated dataset, and therefore

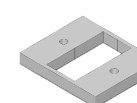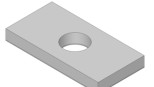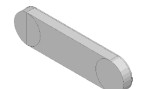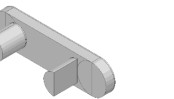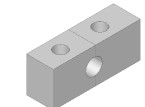

**Basic Caption:**
Fill the rectangular cutout and left circular hole.
**Multi-level Caption:**
Fill cutout, remove one hole, center remaining hole on top.

**Basic Caption:**
Add a cylinder and a rectangular prism to surface.
**Multi-level Caption:**
Attach a cylinder and semi-cylindrical prism to opposite sides.

**Basic Caption:**
Drill three holes into the rectangular prism.
**Multi-level Caption:**
Drill two vertical holes on top, horizontal hole through side.

Figure 9: Comparisons between basic captioning method and our multi-level captioning method.

Table 2: Ablation study on multi-level captioning and selective data. The CAD-Editor-mini is trained on a subset with 10k examples.

| Method | COV ↑ | MMD ↓ | JSD ↓ | D-CLIP ↑ | Valid Ratio ↑ |
|---|---|---|---|---|---|
| CAD-Editor-mini w/ BC | 75.7 | 1.40 | 1.64 | 0.02 | 78.4 |
| CAD-Editor-mini w/ MLC | 78.2 | 1.37 | 1.56 | 0.09 | 88.2 |
| CAD-Editor w/o DCF&HF | 78.5 | 1.19 | 1.50 | 0.11 | 91.8 |
| CAD-Editor w/ DCF | 79.4 | 1.22 | 1.31 | 0.14 | 89.5 |
| CAD-Editor w/ HF | 81.0 | 1.16 | 1.55 | 0.21 | 91.4 |

by the CAD design variation models used to generate the CAD pairs. Furthermore, our method's ability to generalize to new edits and make correct associations between visual changes and text instructions is limited by the ability of GPT-4o to generate instructions.

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

# A  APPENDIX: MULTI-LEVEL CAPTIONING

For multi-level captioning, we utilize GPT-4o four times for each CAD pair to generate the final editing instruction. The detailed prompt is illustrated in Fig 10.

---

**Multi-level Captioning**

*Step 1:*
Please take a look at the first of two 3D shapes we'll be examining. Please provide a detailed description, focusing on its geometric properties, including the type and number of elements it features, the proportions of its size, its positional relationships between elements, and any additional details that stand out.

*Step 2:*
Now, let's turn our attention to the second 3D shape. Please provide a detailed description, focusing on its geometric properties, including the type and number of elements it features, the proportions of its size, its positional relationships between elements, and any additional details that stand out.

*Step 3:*
Please provide detailed instructions for transforming the first 3D shape into the second.

*Step 4:*
Condense your instructions to one sentence, 10 words maximum.

---

Figure 10: The detailed prompt for multi-level captioning.

# B  APPENDIX: BASELINES

Here, we provide the detailed prompt for the baseline method in Figure 11.

---

**GPT-4o (3-shot)**

Modify the original Computer-Aided Design(CAD) operation sequence according to the instruction:
## Instructions for sketch-and-extrude model
A sketch-and-extrude model consists of multiple extruded-sketches.
# Sketch
- A sketch consists of multiple faces
- A face consists of multiple loops.
- A loop consists of multiple curves.
- A curve is either a line, an arc, or a circle.
- A circle is defined by four points with four geometry tokens.
- An arc is defined by three points but with two tokens, where the third point is specified by the next curve (or the first curve when a loop is closed).
- A line is defined by start point.
- A point is represented by two integers which stands for the x and y coordinate, respectively.
- A loop with a circle can not contain additional curves since it is already a closed path.
- When a face consists of multiple loops, the first loop defines the external boundary, and the remaining loops define internal loops (i.e., holes).
- An end-primitive token appears at the end of each primitive (curve, line, face, loop or sketch).
# Extrude
Each sketch will be followed by an extrude, which is represented by 18 parameters: BWVTTTRRRRRRRRRSOO
- B represents one of the three Boolean operations: add, cut or intersect. It occupies 1 parameter
- V indicates the displacements of the top and the bottom planes from the reference plane in which a sketch is extruded to form a solid. It occupies 2 parameters.
- T represents 3D translation applied to the extruded solid. It occupies 3parameters.
- R represents 3D rotation of the extrusion direction. It occupies 6 parameters.
- S represents the uniform scaling factor. It occupies 1 parameter.
- O represents the center of scaling as a 2D coordinate. It occupies 2 parameters.
# Note
- Note that every number is an integer.
## Examples for editing sketch-and-extrude model
[The 3 most similar editing instructions and their corresponding CAD pairs]
## Your task
Original CAD Command Sequence:
[original sequence]
Instruction:
[editing instruction]
Your output should be of the following json format:
{
   "edited sequence": your Modified CAD Command Sequence here.
}

---

Figure 11: The detailed prompt for GPT-4o (3-shot).

# C  APPENDIX: ADDITIONAL RESULTS

Here, we present a qualitative comparison between the initially fine-tuned LLM and the LLM further fine-tuned with selective data in Figure 12. Enhancing fine-tuning with selective data improves generation quality, text-CAD alignment, and output stability.

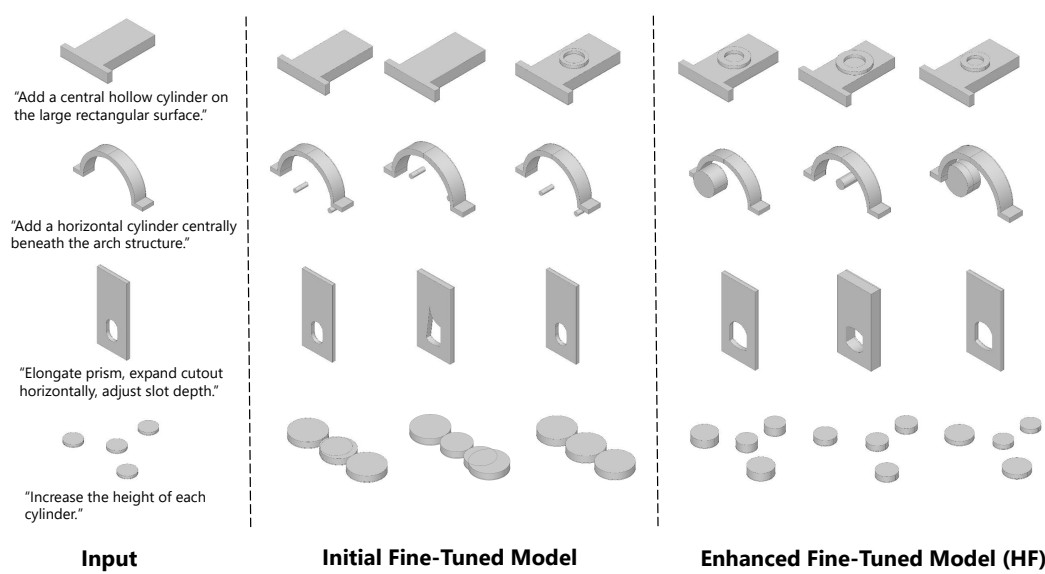

Figure 12: The qualitative comparison between the initial fine-tuned LLM and the enhanced LLM using selective data from human feedback.

More qualitative comparison between CAD-Editor and baselines are shown in Figure 13.

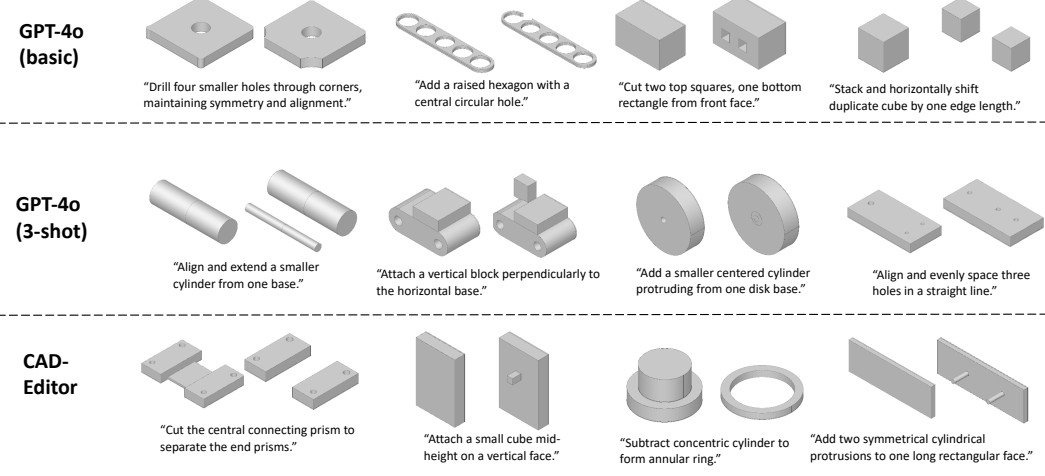

Figure 13: Additional qualitative comparison results between CAD-Editor and baselines.

In addition, CAD-Editor has the ability to identify and correct erroneous editing instructions as shown in Figure 14. For example, it successfully interprets misspellings like "cilynder," incorrect syntax such as "the at bottom," and incomplete words like "remve" and "creat." This capability may stem from its fine-tuning on LLMs, which have strong text comprehension abilities and can effectively handle erroneous text (Cao et al., 2023).

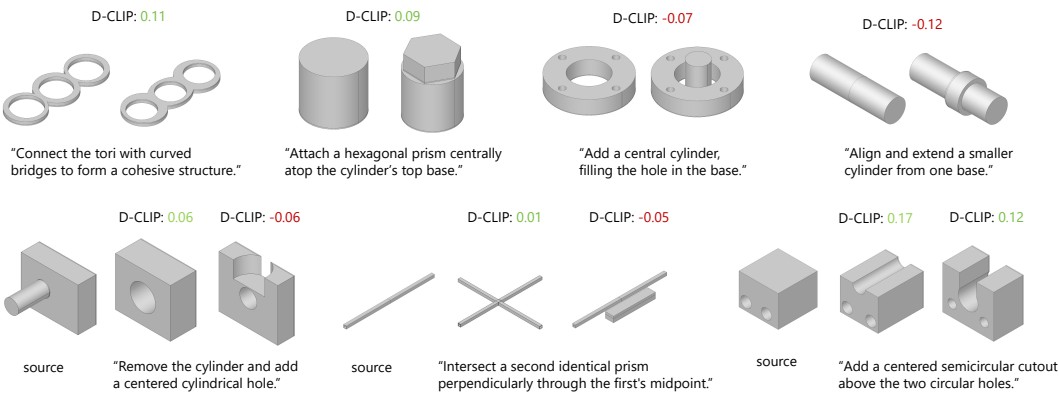

Figure 14: CAD-Editor has the ability to identify and correct erroneous editing instructions.

# D Appendix: Examples of D-CLIP

In this section, we present examples of D-CLIP scores along with their corresponding text and image pairs. As illustrated in Figure 15, D-CLIP quantifies the alignment between the textual editing instructions and the changes observed between the original and edited CAD images.

Figure 15: Examples of D-CLIP scores and their corresponding text and image pairs, demonstrating its ability to evaluate the performance of text-based CAD editing.

# E Appendix: Generalization on Fusion 360 Gallery

To evaluate the generalization capability of our CAD-Editor, we conduct a cross-dataset test. Specifically, we use the model trained on our proposed dataset and test it on a different dataset provided by the Fusion 360 Gallery (Willis et al., 2021b). These datasets originate from different sources: our dataset is based on DeepCAD, which leverages CAD models from the Onshape repository, while the Fusion 360 Gallery dataset is derived from designs created in Autodesk Fusion 360. Using the same data generation process described in Section 3.2, we construct a test set containing 600 samples. For each sample, we perform inference five times, yielding a total of 3,000 results.

As shown in Table E, our CAD-Editor consistently outperforms baseline models across all metrics, demonstrating exceptional quality, diversity, alignment with editing instructions, and validity. Figure 16 further demonstrates its superior qualitative performance. These results highlight the robustness and generalization ability of our approach when applied to datasets with different shape distributions from the training data.

# F Appendix: Fine-tuning Loss on Llama-3 Models.

We conduct additional experiments with Llama-3-70B, and the preliminary result shown in Figure 17 reveals that fine-tuning the 70B model achieves faster loss reduction compared to the 8B model. This suggests that our method has the potential for further improvement with increased computational resources.

Table 3: Quantitative evaluations on dataset generated from Fusion 360 Gallery.

| Method | COV ↑ | MMD ↓ | JSD ↓ | D-CLIP ↑ | Valid Ratio ↑ |
|---|---|---|---|---|---|
| GPT-4o (basic) | 72.9 | 3.05 | 10.9 | -0.58 | 42.8 |
| GPT-4o (3-shot) | 75.1 | 2.40 | 10.2 | 0.00 | 67.9 |
| CAD-Editor | **80.5** | **2.13** | **6.81** | **0.35** | **79.5** |

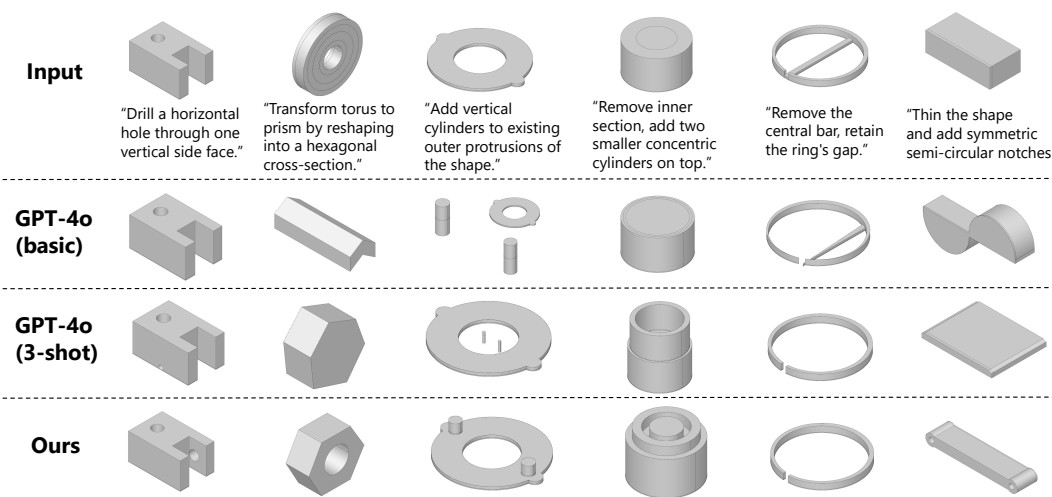

Figure 16: Results for text-based CAD editing by GPT-4o (basic), GPT-4o (3-shot) and CAD-Editor on Fusion 360 Gallery.

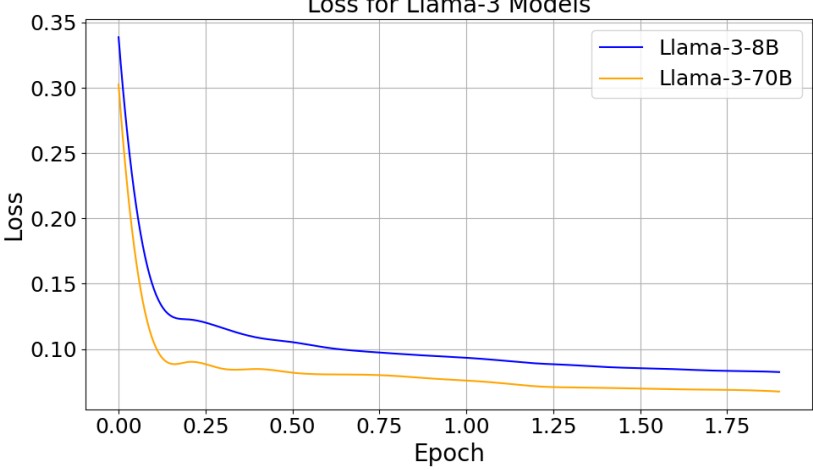

Figure 17: Fine-tuning Loss on Llama-3 Models.

