# OpenReview forum: "CAD-Editor: Text-based CAD Editing through Adapting Large Language Models with Synthetic Data"
_ICLR.cc/2025/Conference — Submitted to ICLR 2025_

### Official Review · Reviewer_CzQp · 2024-10-31

**Soundness:** 3
**Presentation:** 3
**Contribution:** 3
**Rating:** 6
**Confidence:** 3

**Summary:**

The paper proposes a CAD editing dataset. The CAD models are collected and fed into design variation generation models to create design variations. A large language model (LLM) is used to generate multi-level captions, which are then compressed into sentences with a maximum of 10 words. To enhance model performance, the paper suggests filtering some synthetic data using D-CLIP or human evaluation. The model is fine-tuned on LLAMA 3 and achieves the best performance compared to other methods.

**Strengths:**

The paper proposes an interesting dataset focused on instruction-based CAD editing, a task that holds significant value for industrial applications.

The paper is well-written and easy to follow.

It introduces an innovative model-based data selection method that enhances the quality of the dataset.

**Weaknesses:**

I am curious about the D-CLIP evaluation. It is unclear whether the adapted D-CLIP model was pretrained on any specific datasets. Was it fine-tuned on the proposed dataset? If not, is there a potential domain gap between the proposed dataset and the datasets used to train D-CLIP? This domain gap could potentially influence the effectiveness of the data selection process.

**Questions:**

See above.

---

> ### Author Response · Authors · 2024-11-20
> **Response to Reviewer CzQp**
>
> > **Q1.** I am curious about the D-CLIP evaluation. It is unclear whether the adapted D-CLIP model was pretrained on any specific datasets. Was it fine-tuned on the proposed dataset? If not, is there a potential domain gap between the proposed dataset and the datasets used to train D-CLIP? This domain gap could potentially influence the effectiveness of the data selection process.
>
> ------
>
> **A1.** Thank you for your valuable suggestion and we would like to clarify the following points.
>
> (1) D-CLIP is not trained on specific datasets. The training of CLIP requires **pair-wise** data: <a text description, an image>. In our task, the data is in **triplet**: <an editing instruction, CAD image 1, CAD image 2>.
>
> (2) We empirically observe that while there is a domain gap between CAD image and real image used in CLIP pre-training, D-CLIP can evaluate the performance. We add some examples in Appendix D for your reference.

---

> ### Author Response · Authors · 2024-11-25
> **Looking forward to further feedback**
>
> Dear Reviewer CzQp,
>
> Thank you again for your valuable comments and suggestions, which are very helpful to us. In response to your concerns, we have clarified the reasons for not fine-tuning D-CLIP and provided examples in Appendix D to showcase its effectiveness.
>
> We understand that this is quite a busy period, so we sincerely appreciate it if you could take some time to return further feedback on whether our responses resolve your concerns. If there are any other comments, we will try our best to address them.
>
> Best regards,
>
> The Authors

---

> > ### Comment · Reviewer_CzQp · 2024-11-27
> >
> > Thank you for the response. It is still not clear whether the CLIP model is trained on the proposed dataset or some other dataset. If the proposed dataset is also used to train CLIP, how can you ensure that D-CLIP is good enough to evaluate the CAD data? Specifically, there is a concern about the risk of overfitting. If D-CLIP and the LLM model share the same training set, D-CLIP may learn dataset-specific characteristics too finely. This could result in evaluation outcomes that do not generalize well to other data, potentially compromising the reliability and objectivity of the results and also influence the data selection process.

---

> ### Author Response · Authors · 2024-11-29
> **Response to Reviewer CzQp**
>
> Thank you for your insightful questions. We have considered the same questions while developing CAD-Editor. Ultimately, we decided **not** to train CLIP on the proposed dataset, as this approach enhances the effectiveness of evaluation and data selection. The detailed reasons are explained below.
>
> 1. First, we would like to explain how the proposed dataset is built. We start with an existing dataset, **DeepCAD** [1], which consists of **\<CAD model\>**. Then, we leverage design variation models and MLLMs to produce our proposed dataset, **CAD-Editor-Dataset**, which consists of **<textual editing instruction, original CAD model, edited CAD model>**.
>
> 2. We do not fine-tune CLIP using either DeepCAD or CAD-Editor-Dataset. Besides, Neither DeepCAD nor CAD-Editor-Dataset were included in the training set of CLIP.
>
>    (1) DeepCAD is available on the Internet as **JSONs rather than images**, whereas CLIP is trained on images crawled from the Internet paired with corresponding alt-texts.
>
>    (2) CAD-Editor-Dataset is **largely synthetic** and **preprocessed using our unique method**, making it entirely new. Additionally, while CLIP is trained on pair-wise data (<text description, image>), our dataset consists of triplets (<textual editing instruction, original CAD model, edited CAD model>). These make it unlikely to be part of CLIP's training data.
>
> 3. There are **CAD-related data in CLIP's training set**. We have searched LAION (the dataset used to train CLIP), and found images related to CAD designs, but they are not identical to DeepCAD.
>
> Based on the above investigation about DeepCAD, CAD-Editor-Dataset and CLIP’s training data, we have the following discussion about the effectiveness of evaluation and data selection.
>
> 1. Since CLIP was not trained on the DeepCAD or CAD-Editor-Dataset, there is **no concern regarding overfitting** during evaluation. Furthermore, since CLIP has been exposed to some CAD-related images (but not exactly from DeepCAD and CAD-Editor-Dataset) and has been trained on an extensive dataset (approximately 400 million), it possesses **strong generalization capabilities**. This makes D-CLIP efficient for evaluation and data selection. We also provide examples in Appendix D to illustrate this point.
>
> 2. This aligns with research on text-to-image tasks [2, 3, 4, 5], where CLIP or D-CLIP is used as a metric for similar reasons: CLIP was pre-trained on the data related to the target task but not fine-tuned on the exact dataset of the target task.
>
> [1] Wu R, et al. DeepCAD: A Deep Generative Network for Computer-Aided Design Models[C]. International Conference on Computer Vision (ICCV), 2021.
>
> [2]  Qu L, et al. Discriminative probing and tuning for text-to-image generation[C]. Proceedings of the IEEE/CVF Conference on Computer Vision and Pattern Recognition (CVPR), 2024.
>
> [3] Saharia C, et al. Photorealistic text-to-image diffusion models with deep language understanding[C]. Advances in Neural Information Processing Systems (NeurIPS), 2022.
>
> [4] Gal R, et al. StyleGAN-NADA: CLIP-guided domain adaptation of image generators[J]. ACM Transactions on Graphics (TOG), 2022.
>
> [5] Brooks T, et al. InstructPix2Pix: Learning to Follow Image Editing Instructions[C]. Proceedings of the IEEE/CVF Conference on Computer Vision and Pattern Recognition (CVPR), 2023.

---

### Official Review · Reviewer_F1Vw · 2024-11-02

**Soundness:** 3
**Presentation:** 3
**Contribution:** 2
**Rating:** 5
**Confidence:** 4

**Summary:**

The paper introduces CAD-Editor, a text-driven CAD editing method based on large language models (LLMs). To generate high-quality training data, the article adopts a two-stage training strategy. The first stage generates a large-scale synthetic dataset, and the second stage constructs a selective dataset. Experimental results show that CAD-Editor outperforms other methods on multiple evaluation metrics, including GPT-4o.

**Strengths:**

- Innovatively transforms CAD editing tasks into sequence-to-sequence problems, leveraging large language models (LLMs) to achieve text-driven CAD model editing.
- Adopts a multi-stage training strategy, significantly improving model performance by combining synthetic datasets and selective datasets.

**Weaknesses:**

- I am deeply concerned about the generalization ability of CAD-Editor. On one hand, the choice of the base dataset is crucial for the model's generalization ability, and CAD-Editor is only optimized on the DeepCAD dataset. On the other hand, the generation of the synthetic dataset relies on existing CAD generation models (such as SkexGen and Hnc-CAD), both of which are also trained on DeepCAD. Therefore, the performance of CAD-Editor on the DeepCAD benchmark raises suspicions of overfitting. That is, if the fine-tuning dataset mainly contains CAD models from a specific industry (such as the automotive industry), the trained model may not handle CAD models from other industries (such as aerospace or construction) well.
- The reliance on GPT-4o. The proposed method requires multiple prompts to GPT-4o to obtain generated data. The article does not discuss the token cost of GPT-4o and potential performance bottlenecks.

**Questions:**

- The article mainly focuses on models with 8B parameters, and based on the qualitative analysis results provided, there is still a significant gap between the current model's effectiveness and being able to provide productivity. I am curious whether CAD-Editor can be more effective when scaled to larger models or more diverse datasets.
- How does CAD-Editor perform when dealing with vague or ambiguous text instructions? Does the model have the ability to identify and correct erroneous editing instructions?

---

> ### Author Response · Authors · 2024-11-20
> **Response to Reviewer F1Vw: Part (1)**
>
> > **Q1.** I am deeply concerned about the generalization ability of CAD-Editor. On one hand, the choice of the base dataset is crucial for the model's generalization ability, and CAD-Editor is only optimized on the DeepCAD dataset. On the other hand, the generation of the synthetic dataset relies on existing CAD generation models (such as SkexGen and Hnc-CAD), both of which are also trained on DeepCAD. Therefore, the performance of CAD-Editor on the DeepCAD benchmark raises suspicions of overfitting. That is, if the fine-tuning dataset mainly contains CAD models from a specific industry (such as the automotive industry), the trained model may not handle CAD models from other industries (such as aerospace or construction) well.
>
> ------
>
> **A1.**  Thank you for your feedback.
>
> (1) **Related works on generating CAD models are predominantly trained on the DeepCAD [1, 2, 3], which is a standard in this field.** To our knowledge, no other datasets comparable to DeepCAD are available. There are two primary reasons for selecting this dataset. First, DeepCAD provides **a substantial amount of data** for training purposes (approximately 178k instances), whereas other datasets offer significantly less data (for instance, Fusion 360 [4] contains only about 8k instances). Second, DeepCAD includes **sketch-extrusion histories**, while other datasets typically contain only B-Rep or mesh information.  Third, DeepCAD is derived from the ABC dataset, which contains a good sampling of **diverse shapes in different industries** [5].
>
> (2) **Studies have indicated that using DeepCAD is unlikely to result in overfitting** due to its extensive and cross-industry data, and during data collection, no specific domains were excluded. According to the research that introduces DeepCAD [3] (supplementary E), the model trained on DeepCAD generalize well to Fusion 360, which is collected from different sources to DeepCAD.
>
> **We also conduct similar experiments to demonstrate that CAD-Editor has great generalization ability (see Appendix E).** Specifically, we use the model trained on DeepCAD to directly perform inference on Fusion 360. Experiments show that our CAD-Editor not only surpasses the baseline models in quantitative results but also delivers visually superior results in qualitative evaluations, confirming the generalization ability of our approach when applied to datasets with shape distributions that differ from the training dataset.
>
> (3) Regarding whether a model trained on CAD from one industry, such as the automotive industry, can effectively handle CAD from vastly different industries, like aerospace or construction, we consider this **an open problem**. Specifically, our opinion is that transferring knowledge between closely related industries, as discussed in A1 point (2), is feasible. However, transferring knowledge from one industry to a vastly different one presents considerable challenges.
>
> It is also an open problem for the entire AI community. Similar questions arise in other domains. For instance, in the NLP domain, an analogous question is whether a model trained solely on English (paralleling the automotive industry in our context) can generate Japanese (representing the aerospace or construction industries in our context), given that both English and Japanese are human languages.
>
> We appreciate discussing this issue but note it is not the focus of our work. **Our goal is to develop the first model for text-based CAD editing. Experiments on DeepCAD, a common and standard dataset in CAD generation studies, show our method's effectiveness.**
>
> [1] Wu R, et al. DeepCAD: A Deep Generative Network for Computer-Aided Design Models. International Conference on Computer Vision (ICCV), 2021.
>
> [2] Xu X, et al. SkexGen: Autoregressive Generation of CAD Construction Sequences with Disentangled Codebooks[C]. International Conference on Machine Learning (ICML), 2022.
>
> [3] Xu X, et al. Hierarchical Neural Coding for Controllable CAD Model Generation[C]. International Conference on Machine Learning (ICML), 2023.
>
> [4] Willis K D D, et al. Fusion 360 gallery: A dataset and environment for programmatic cad construction from human design sequences[J]. ACM Transactions on Graphics (TOG), 2021.
>
> [5] Koch S, et al. ABC: A big cad model dataset for geometric deep learning[C]. Proceedings of the IEEE/CVF conference on computer vision and pattern recognition (CVPR), 2019.

---

> ### Author Response · Authors · 2024-11-20
> **Response to Reviewer F1Vw: Part (2)**
>
> > **Q2.** The reliance on GPT-4o. The proposed method requires multiple prompts to GPT-4o to obtain generated data. The article does not discuss the token cost of GPT-4o and potential performance bottlenecks.
>
> ------
>
> **A2.** Thank you for your feedback.
>
> (1) The detailed prompts we used are provided in Appendix A and B. To construct each data, the average input is 485 tokens. We use Azure OpenAI API, where the quota limit in tokens per minute (TPM) of GPT-4o is quite large (i.e., 450k), which is **enough to support our experiment**. We also find that the **very recent LLaVA-OneVision could be an alternative** to GPT-4o for researchers with budget constraints.
>
> (2) Other potential performance bottlenecks may arise from GPT-4o's ability to accurately  recognize and describe transformations, such as incorrect positional relationships, numbers, or shape types. To address this, we have introduced a **Multi-level Captioning** strategy that breaks down this complex task into several simpler sub-tasks, thereby improving the overall accuracy. Details can be found in lines 241-257. Moreover, we have shown comparisons between basic captioning method and our multi-level captioning method in Figure 9.
>
>
> > **Q3.** The article mainly focuses on models with 8B parameters, and based on the qualitative analysis results provided, there is still a significant gap between the current model's effectiveness and being able to provide productivity. I am curious whether CAD-Editor can be more effective when scaled to larger models or more diverse datasets.
>
> ------
>
> **A3.**  Thank you for your insightful suggestion.
>
> (1) Evidence in related work shows that larger models usually lead to better performance [1, 2].
>
> (2) Due to limited computational resources and time constraints, we were unable to explore this scaling effect on CAD data before. However, we are trying to conduct additional experiments using Llama-3-70B to investigate this further. As this experiment requires significant time, we can only share preliminary results during the Author/Reviewer Discussion in Appendix F. **Initial findings show that fine-tuning the 70B model achieves a faster loss reduction compared to the 8B model**, suggesting that our method has the potential for further improvement with increased computational resources.
>
> (3) Regarding the potential of using more diverse datasets, we have addressed this topic in A1.
>
> [1] Gruver N, et al. Fine-Tuned Language Models Generate Stable Inorganic Materials as Text[C]. The Twelfth International Conference on Learning Representations (ICLR), 2024.
>
> [2] Yu L, et al. MetaMath: Bootstrap Your Own Mathematical Questions for Large Language Models[C]. The Twelfth International Conference on Learning Representations (ICLR), 2024.
>
>
>
> > **Q4.** How does CAD-Editor perform when dealing with vague or ambiguous text instructions? Does the model have the ability to identify and correct erroneous editing instructions?
>
> ------
>
> **A4.** Thank you for your constructive suggestion.
>
> (1) **We have shown in Figure 7 that CAD-Editor handles vague or ambiguous text instructions well.** For example, when given the instruction "Add smaller prisms on the top",  CAD-Editor generates a variety of valid interpretations, such as a hexagonal prism on top of a base rectangle, an 'X' prism, two triangular prisms, two trapezoidal prisms, or four rectangular prisms. The results vary in shape, quantity, and horizontal positioning, demonstrating the model's flexibility and creativity in interpreting and executing vague or ambiguous instructions.
>
> (2) **CAD-Editor can identify and correct errors in instructions and we have added examples in Figure 14, Appendix C**. For example,  it successfully interprets misspellings like "cilynder," incorrect syntax such as "the at bottom," and incomplete words like "remve" and "creat." This capability may stem from its fine-tuning on LLMs, which have strong text comprehension abilities. Recent studies [1] have shown that LLMs can effectively handle erroneous text .
>
> [1] Cao Q, et al. Unnatural error correction: GPT-4 can almost perfectly handle unnatural scrambled text[C]. Proceedings of the 2023 Conference on Empirical Methods in Natural Language Processing (EMNLP), 2023.

---

> ### Author Response · Authors · 2024-11-25
> **Looking forward to further feedback**
>
> Dear Reviewer F1Vw,
>
> Thank you again for your valuable comments and suggestions, which are very helpful to us. We have provided responses to the concerns raised.
>
> 1. We have explained that DeepCAD, with its cross-industry diversity and substantial size, is a robust choice for training. Additionally, we have conducted additional experiments to demonstrate CAD-Editor's ability to generalize to unseen distributions, the results are presented in Appendix E.
> 2. Our experiments show that the token quota of GPT-4o is sufficient. To address other potential bottlenecks of GPT4-o, our Multi-level Captioning strategy improves performance.
> 3. Scaling to larger models has been proved effective in related works. We have conducted additional experiments with Llama-3-70B and our preliminary results show faster fine-tuning convergence.
> 4. CAD-Editor effectively handles vague or ambiguous instructions as shown in Figure 7. We also perform additional experiments to show that CAD-Editor can identify and correct erroneous editing instructions (see Figure 14 in Appendix C for examples).
>
> We understand that this is quite a busy period, so we sincerely appreciate it if you could take some time to return further feedback on whether our responses resolve your concerns. If there are any other comments, we will try our best to address them.
>
> Best regards,
>
> The Authors

---

> > ### Comment · Reviewer_F1Vw · 2024-11-26
> >
> > Sorry for the late reply. I appreciate the authors’ efforts in responding to comments. My concerns have been largely addressed. Additionally, I hope that, if possible, the authors can provide a comprehensive discussion with a similar work [1].
> >
> > [1] Text2CAD: Generating Sequential CAD Designs from Beginner-to-Expert Level Text Prompts. NeurIPS 2024, Spotlight.

---

> ### Author Response · Authors · 2024-11-26
> **Response to Reviewer F1Vw**
>
> Thank you very much for your reply, which encourages us greatly!
>
> Text2CAD was recently published on arXiv on September 25, 2024. Due to the submission deadline for ICLR being October 1, 2024, we did not have adequate time to include a discussion of Text2CAD in our submitted version. We provide a comprehensive discussion of Text2CAD here and will incorporate it into the revised version.
>
> ## **1. Task**
>
> Both works introduce new tasks in CAD but tackle fundamentally different objectives:
>
> - **Text2CAD**: Focuses on **text-based CAD generation**, where the goal is to generate CAD models from natural language descriptions. Given an input text, Text2CAD produces a corresponding CAD sequence.
> - **Our Work**: Focuses on **text-based CAD editing**, where the objective is to modify CAD models based on textual instructions. CAD-Editor takes both the original CAD sequence and the textual instruction as input and outputs the edited CAD sequence.
>
> ## **2. Data Generation**
>
> Both works generate new datasets based on DeepCAD but differ in the following aspects.
>
> ### **(1) Data Formatting**
>
> - **Text2CAD**: Generates **pairwise data** in the form of < text description, CAD model>.
> - **Our Work**: Generates **triplet data** in the form of <editing instruction, original CAD model, edited CAD model>, which is tailored for the editing task.
>
> ### **(2) CAD Sequence Representation**
>
> - **Text2CAD**: Follows the CAD sequence representation from CAD-SIGNet [1].
>
> - **Our Work**: Adapts the representation from SkexGen [2], with improvements for **better readability and flexibility**.
>
>   **Main differences:**
>
>   - We use **text tokens** (natural language descriptions) instead of binary representations for categorical operations (e.g., curve types, extrusion operations) and numerical attributes (e.g., point coordinates, extrusion parameters). This enhances readability and comprehensibility.
>   - We support **variable token lengths**, avoiding placeholders to standardize command lengths. This simplifies processing and reduces sequence length, making it more efficient for LLMs to interpret.
>
> ### **(3) Data Generation Methodology**
>
> - **Text2CAD**: Generates text prompts describing the construction workflow of a CAD model with **varying complexities**.
>
>   1. **Shape description generation using VLM**: Generate abstract object-level descriptions of the CAD models.
>   2. **Multi-Level textual annotation generation using LLM**: Generate multiple textual annotations corresponding to different design details of a CAD model through middle-column.
>
> - **Our Work**: Generates CAD model pairs and corresponding **accurate and high-quality** textual editing instructions:
>
>   1. **Paired CAD Generation via Design Variation**: Generate CAD pairs by feeding existing CAD models into design variation models to produce their variants.
>   2. **Textual Instruction Generation via Multi-Level Captioning**: Render CAD model pairs into images and use MLLMs to extract textual differences. We introduce multi-level captioning strategy which breaks the task into three simpler sub-tasks: (1) Describe CAD; (2) Identify differences (3) Compress instructions. This improves the accuracy and quality of the generated instructions.
>
>   Note that the term "multi-level" has different meanings in two works. In Text2CAD, it refers to varying complexities of the generated text prompts. In our work, it refers to breaking down instruction generation into simpler sub-tasks.
>
> ## **3. Model and Training**
>
> - **Text2CAD**: Proposes an **end-to-end transformer-based autoregressive network**.
>
> - **Our Work**: Leverages a **pre-trained LLM** as the backbone, enhanced via a **two-stage fine-tuning process**: (1) initial fine-tuning with synthetic data; (2) enhanced fine-tuning with selective data.
>
>   **Advantages of Our Approach**
>
>   **(1) Inherits the strong capabilities of LLMs:**
>
>   - **Strong Text Comprehension**: LLMs inherently possess strong text comprehension abilities, which are beneficial in handling diverse and complex textual instructions.
>   - **CAD Knowledge in Pre-trained LLMs**: LLMs have learned CAD-related codes in the pre-training [3], aiding in understanding CAD’s SE operations and geometric concepts.
>   - **Robustness to Vague or Erroneous Instructions**: LLMs excel in interpreting ambiguous or erroneous instructions, ensuring robustness in real-world scenarios.
>
>   **(2) Utilizes Data Practically and Efficiently:** The combination of synthetic and selective data offers a practical and efficient way to obtain sufficient training data when naturally existing datasets are absent.
>
> [1] Khan M S, et al. CAD-SIGNet: CAD Language Inference from Point Clouds using Layer-wise Sketch Instance Guided Attention. CVPR 2024.
>
> [2] Xu X, et al. SkexGen: Autoregressive Generation of CAD Construction Sequences with Disentangled Codebooks. ICML 2022.
>
> [3] Makatura L, et al. How Can Large Language Models Help Humans in Design and Manufacturing?. arXiv preprint arXiv:2307.14377, 2023.

---

> ### Author Response · Authors · 2024-11-29
> **Kindly Request Further Feedback**
>
> Dear Reviewer F1Vw,
>
> We sincerely appreciate the time and efforts you have dedicated to reviewing our work. Your expert opinions and constructive feedback are crucial in helping us improve the quality of our work.
>
> Following your suggestion, we have provided a comprehensive discussion with Text2CAD. It is important for us to confirm whether our responses have adequately addressed your concerns. We would be very grateful if you could provide any further comments at your earliest convenience.
>
> We look forward to your valued feedback and thank you once again for your careful consideration.
>
> Best regards,
>
> The Authors

---

### Official Review · Reviewer_Bbso · 2024-11-04

**Soundness:** 2
**Presentation:** 3
**Contribution:** 2
**Rating:** 5
**Confidence:** 4

**Summary:**

The paper introduces CAD-Editor, a novel approach for text-based CAD editing that leverages LLMs to interpret and apply textual instructions to CAD models. CAD-Editor is designed as a sequence-to-sequence generation model, enabling users to edit CAD models via natural language inputs. The paper outlines a method involving two phases of fine-tuning on synthetic and selective datasets to optimize performance. Experimental results demonstrate CAD-Editor’s improvements in accuracy, alignment with textual instructions, and CAD model validity over other models, including SkexGen and Hnc-CAD. This work also establishes a benchmark for text-based CAD editing.

**Strengths:**

- The paper is well-written and it's easy to follow.
- As the authors have also mentioned, CAD generation and editing has gotten relatively little research attention although they play a huge role in different industries.

**Weaknesses:**

- I believe this paper is a better fit for a computer vision conference for being more of an applied paper investigating CAD editing.
- I think the authors should provide a couple of examples of D-CLIP scores and their corresponding text/image pairs to show the effectiveness of their proposed metric.

**Questions:**

- In line 69, what do the authors mean by "without control over the final appearance"? Aren't they text to CAD models?
- In line 259, how do you automatically exclude instructions with more than three edits?
- Have you tried fine-tuning any of the baseline text-to-cad models instead of LLama3-8b-instruct?

---

> ### Author Response · Authors · 2024-11-20
> **Response to Reviewer Bbso**
>
> > **Q1.** I believe this paper is a better fit for a computer vision conference for being more of an applied paper investigating CAD editing.
>
> ***
>
> **A1.** Thank you for your feedback. Our paper focuses on CAD editing and LLM, aligning well with the topic of leading machine learning conferences such as ICLR, ICML, and NeurIPS, which **regularly feature work at the intersection of machine learning and computer vision**. For example, ICML has published two works we cited in our paper -- SkexGen [1] and Hnc-CAD [2]. This year, NeurIPS included Text2CAD [3], and previous ICLR conferences have presented related studies [4, 5, 6].
>
> [1] Xu X, et al. SkexGen: Autoregressive Generation of CAD Construction Sequences with Disentangled Codebooks[C]. International Conference on Machine Learning (ICML), 2022.
>
> [2] Xu X, et al. Hierarchical Neural Coding for Controllable CAD Model Generation[C]. International Conference on Machine Learning (ICML), 2023.
>
> [3] Khan M S, et al. Text2CAD: Generating Sequential CAD Models from Beginner-to-Expert Level Text Prompts[J]. Conference on Neural Information Processing Systems (NeurIPS), 2024.
>
> [4] Seff A, et al. Vitruvion: A Generative Model of Parametric CAD Sketches[C].10th International Conference on Learning Representations (ICLR), 2022.
>
> [5] Salihu D, et al. DeepSPF: Spherical SO (3)-Equivariant Patches for Scan-to-CAD Estimation[C]. The Twelfth International Conference on Learning Representations (ICLR), 2024.
>
> [6] Lei H, et al. CircNet: Meshing 3D Point Clouds with Circumcenter Detection[C]. The Eleventh International Conference on Learning Representations (ICLR), 2023.
>
>
>
> > **Q2.** I think the authors should provide a couple of examples of D-CLIP scores and their corresponding text/image pairs to show the effectiveness of their proposed metric.
>
> ------
>
> **A2.**   Thank you for your suggestion. We've **added examples of D-CLIP scores with their corresponding text/image pairs in Appendix D**. Besides, **D-CLIP has been shown to be effective in other domains before we adapt it for CAD**. In lines 290-293, we have said that D-CLIP is initially proposed to measure whether the changes between two images align with the changes between two texts in the CLIP space for image domain adaption problem, and it has been shown to be effective in StyleGAN-NADA [1] ( Section 5.4). D-CLIP has also been used in previous editing works for quantitative comparisons [2]. We adapted D-CLIP for our text-based CAD editing task, with details provided in lines 297-302.
>
> [1] Gal R, et al. Stylegan-nada: Clip-guided domain adaptation of image generators[J]. ACM Transactions on Graphics (TOG), 2022.
>
> [2] Brooks T, et al. Instructpix2pix: Learning to follow image editing instructions[C]. Proceedings of the IEEE/CVF Conference on Computer Vision and Pattern Recognition (CVPR), 2023.
>
>
>
> > **Q3.** In line 69, what do the authors mean by "without control over the final appearance"? Aren't they text to CAD models?
>
> ------
>
> **A3.** **Our work is the first to apply text control to CAD models**. Previous methods primarily focused on **randomly generating or editing** certain parts of a CAD model **without using text input** [1, 2, 3]. As we have described in lines 129-135, DeepCAD achieves random CAD generation [1]. SkexGen [2] allows variations in a CAD model’s topology, geometry, or extrusion level through disentangled codebooks. Hnc-CAD [3] enables variations at loop, profile, or solid levels, completing an entire CAD model from a partial one using a neural code tree representation.
>
> [1] Wu R, et al. DeepCAD: A Deep Generative Network for Computer-Aided Design Models. International Conference on Computer Vision (ICCV), 2021.
>
> [2] Xu X, et al. SkexGen: Autoregressive Generation of CAD Construction Sequences with Disentangled Codebooks[C]. International Conference on Machine Learning (ICML), 2022.
>
> [3] Xu X, et al. Hierarchical Neural Coding for Controllable CAD Model Generation[C]. International Conference on Machine Learning (ICML), 2023.
>
>
>
>
> > **Q4.** In line 259, how do you automatically exclude instructions with more than three edits?
>
> ------
>
> **A4.** We examine the generated data and find that instructions generally use periods ('.') or semicolons (';') as separators. Therefore, we filter out instructions containing more than two of these separators.
>
>
>
>
> > **Q5.** Have you tried fine-tuning any of the baseline text-to-cad models instead of LLama3-8b-instruct?
>
> ------
>
> **A5.** Thank you for your suggestion. However, **there were no existing text-to-CAD models available** for fine-tuning, as our work is the first to apply text control to CAD models.

---

> ### Author Response · Authors · 2024-11-25
> **Looking forward to further feedback**
>
> Dear Reviewer Bbso,
>
> Thank you again for your valuable comments and suggestions, which are very helpful to us. We have provided responses to the concerns raised.  First, we have explained the suitability of our work for ICLR. Second, we have added examples of D-CLIP scores with their corresponding text/image pairs to show the effectiveness in Appendix D as suggested. Third, regarding Q3 and Q5, we clarified potential misunderstandings, as our work is the first to apply text-based control to CAD models. Fourth, we also provided a clear explanation of how instructions are automatically filtered.
>
> We understand that this is quite a busy period, so we sincerely appreciate it if you could take some time to return further feedback on whether our responses resolve your concerns. If there are any other comments, we will try our best to address them.
>
> Best regards,
>
> The Authors

---

> > ### Comment · Reviewer_Bbso · 2024-11-27
> >
> > I would like to thank the authors for their response.
> >
> > I am aware that there are papers on CV and CAD in top-tier ML conferences. My comment was meant to express that while the contribution seems somewhat limited, the application might be more compelling in CV-focused conferences.
> >
> > Some of the concerns I raised have been addressed in their reply. As a result, I am increasing my score; however, I still believe the paper falls below the acceptance threshold.

---

> ### Author Response · Authors · 2024-11-29
> **Response to Reviewer Bbso**
>
> Thank you very much for your thoughtful reply and reassessment of the rating! We appreciate the opportunity to further clarify our work's contributions and suitability for ICLR.
>
> (1) The major contributions of our work are as follows:
>
> 1. **Novel Task**: We introduce a new task called text-based CAD editing. Reviewer CzQp noted that it “holds significant value for industrial applications,” while Reviewer Bbso highlighted that it addresses a research area with “relatively little research attention, despite its importance.”
>
> 2. **Innovative Problem Formulation**: We formulate text-based editing as a seq2seq problem and employ pre-trained LLMs as our backbone. Reviewer F1Vw described this approach as "innovative transformation of CAD editing tasks into sequence-to-sequence problems, utilizing LLMs for text-driven CAD model editing."
>
> 3. **Innovative Training Strategy and superiority of CAD-Editor:** We propose CAD-Editor, the first model specifically for text-based CAD editing. Our multi-stage training strategy, including initial fine-tuning with synthetic data and enhanced fine-tuning with selective data, significantly improves performance. Reviewer F1Vw praised the "multi-stage training strategy, significantly improving model performance by combining synthetic and selective datasets," and Reviewer CzQp highlighted our "innovative model-based data selection method that enhances dataset quality."
>
> 4. **New Benchmark Dataset**: We introduce the first benchmark for text-based CAD editing utilizing design variation model and MLLMs. Our proposed Multi-level Captioning method enhances captioning accuracy. Reviewer CzQp noted it as "an interesting dataset focused on instruction-based CAD editing".
>
> 5. **Clarity**: Both Reviewer Bbso and Reviewer CzQp appreciated that the paper is "well-written and easy to follow."
>
> (2) A very recent work [1], which focuses on another novel task in CAD - text-based CAD generation - has been accepted as a spotlight at NeurIPS 2024. This suggests that such tasks align well with the focus of leading machine learning conferences like ICLR, ICML, and NeurIPS.
>
> We believe that our contributions represent a meaningful advancement in the field of CAD and provide a new perspective for LLM fine-tuning. We would be truly grateful if the additional clarifications provided, along with the strengths of our work could lead to a favorable reassessment above the acceptance threshold.
>
> [1] Khan M S, et al. Text2CAD: Generating Sequential CAD Models from Beginner-to-Expert Level Text Prompts[C]. Conference on Neural Information Processing Systems (NeurIPS), 2024.

---

### Author Response · Authors · 2024-12-02
**Global Response by Authors**

We would like to thank all reviewers for providing constructive feedback that help us improve the paper. We look forward to your further feedback.

We are encouraged that reviewers think our paper:

1. **Novel Task Introduction**: We introduce a new task called text-based CAD editing. Reviewer CzQp noted that it "holds significant value for industrial applications", while Reviewer Bbso highlighted that it addresses a research area with "relatively little research attention, despite its importance".

2. **Innovative Problem Formulation**: We formulate text-based editing as a seq2seq problem and employ pre-trained LLMs as our backbone. Reviewer F1Vw described this approach as "innovative transformation of CAD editing tasks into sequence-to-sequence problems, utilizing LLMs for text-driven CAD model editing."

3. **Innovative Training Strategy and superiority of CAD-Editor**: We propose CAD-Editor, the first model specifically for text-based CAD editing. Our multi-stage training strategy, combining synthetic and selective datasets, significantly enhances performance. Reviewer F1Vw praised the "multi-stage training strategy, significantly improving model performance by combining synthetic and selective datasets", and Reviewer CzQp highlighted our "innovative model-based data selection method that enhances dataset quality."

4. **New Benchmark Dataset**: We introduce the first benchmark for text-based CAD editing utilizing design variation model and MLLMs. Our proposed Multi-level Captioning method enhances captioning accuracy. Reviewer CzQp noted it as "an interesting dataset focused on instruction-based CAD editing".

5. **Clarity**: Both Reviewer Bbso and Reviewer CzQp appreciated that the paper is "well-written and easy to follow."

In our rebuttal, we addressed all raised concerns and misunderstandings:

1. **Misunderstandings and Explanations:**
   - For alignment with ICLR raised by Reviewer Bbso, we explained that our work is highly relevant to major machine learning conferences like ICLR, ICML, NeurIPS, aligning well with their focus and precedent publications.
   - Regarding whether fine-tuning existing text-to-CAD models was considered raised by Reviewer Bbso, we clarified that there were no existing text-to-CAD models available for fine-tuning, as our work is the first to apply text control to CAD models.
   - Addressing concerns about GPT-4o's token costs and bottlenecks raised by Reviewer F1Vw, we noted the manageable token usage and our Multi-level Captionning method which improve performance, and suggested alternatives like LLaVA-OneVision for those with budget constraints.
   - Following Reviewer F1Vw's suggestion, we provided a comprehensive discussion with a recent work Text2CAD, which was published on September 25, 2024 and received by NeurIPS 2024 as a spotlight.

2. **Additional Experiments:**
   - For CAD-Editor's potential overfitting to the DeepCAD dataset raised by Reviewer F1Vw, we clarify that its substantial size and broad scope across industries makes overfitting less likely. Moreover, empirical evidence supports its ability to generalize, as demonstrated in related works. Furthermore, we conducted additional experiments on the Fusion 360 dataset, showcasing CAD-Editor’s robust generalization capabilities (details in Appendix E).
   - Regarding whether CAD-Editor can be more effective when scaled to larger models raised by Reviewer F1Vw, our preliminary results (see Appendix F) show that fine-tuning the 70B model achieves a faster loss reduction compared to the 8B model.
   - For the ability of our CAD-Editor to deal with vague and erroneous instructions, we have shown in Figure 7 that CAD-Editor handles vague or ambiguous text instructions well. Moreover, we added additional experiments to show CAD-Editor can identify and correct errors in instructions (see Figure 14, Appendix C).
   - We addressed Reviewer Bbso's request by adding D-CLIP score examples in Appendix D and responded to Reviewer CzQp's concerns about D-CLIP by clarifying that since CLIP was not trained on the DeepCAD or CAD-Editor-Dataset, overfitting during evaluation is not a concern. We highlighted CLIP’s training on a vast dataset of 400 million images which included CAD-related content, underscoring its robust generalization abilities.

Please see our reviewer-specific feedback for more information.

---

### Meta-Review · Area_Chair_yF4U · 2024-12-22

**Metareview:**

This paper introduces CAD-Editor, an approach leveraging LLMs for text-based CAD editing by framing it as a sequence-to-sequence problem. The methodology includes a multi-stage training strategy with synthetic and selective datasets and introduces a tailored benchmark dataset.  The paper has notable strengths. Text-based CAD editing addresses a significant area with practical industrial implications, and the innovative use of LLMs to interpret and apply textual instructions to CAD models is a promising step forward, likely to inspire further research. Additionally, the paper is well-written and effectively communicates complex ideas.

However, concerns remain that prevent acceptance in its current form. A key issue is the model's ability to generalize beyond the specific datasets used for training and evaluation. The reliance on these datasets raises questions about the applicability of the approach to CAD models from other domains. Furthermore, the computational demands and dependence on large-scale models like GPT-4 pose challenges regarding the practicality and scalability of the method in real-world settings. While the authors have attempted to address some of these issues, they remain significant and highlight areas that require further refinement and validation.

While the paper presents innovative ideas with significant potential, it requires further development to address key concerns. Enhancing generalizability, tackling scalability challenges, and providing more robust validation of the approach are necessary to strengthen its contributions. As such, this paper does not meet the requirements for acceptance in its current form, but I encourage the authors to refine their work for future consideration.

**Additional Comments On Reviewer Discussion:**

During the rebuttal period, the reviewers raised important concerns about the paper's generalizability, computational demands, and evaluation methods. The reviewers raised various concerns about the paper. Reviewer Bbso questioned its fit for the conference and its limited contribution, to which the authors responded by emphasizing its relevance and providing additional examples and clarifications. Reviewer F1Vw highlighted concerns about the model’s ability to generalize beyond the DeepCAD dataset, potential overfitting, and reliance on GPT-4 due to computational costs and scalability issues. The authors conducted additional experiments to address these points. Reviewer CzQp acknowledged the paper’s contributions but raised questions about the D-CLIP evaluation metric, citing potential domain gaps and overfitting risks if D-CLIP and the model shared training data. In response, the authors provided examples to demonstrate D-CLIP’s effectiveness in their context.

In weighing these points, the current version still faces significant concerns. The model's generalizability remains unconvincing, and the evaluation methods require further refinement. The authors are encouraged to address these critical aspects—enhancing generalization studies, addressing computational challenges, and strengthening evaluation methods—for future submissions.

---

### Decision · Program_Chairs · 2025-01-22

Reject